# Composition of Sugars, Organic Acids, Phenolic Compounds, and Volatile Organic Compounds in Lingonberries (*Vaccinium vitis-idaea* L.) at Five Ripening Stages

**DOI:** 10.3390/foods12112154

**Published:** 2023-05-26

**Authors:** Mathias Amundsen, Anne Linn Hykkerud, Niina Kelanne, Sanni Tuominen, Gesine Schmidt, Oskar Laaksonen, Baoru Yang, Inger Martinussen, Laura Jaakola, Kjersti Aaby

**Affiliations:** 1Nofima AS, Osloveien 1, 1340 Ås, Norway; gesine.schmidt@nofima.no (G.S.); kjersti.aaby@nofima.no (K.A.); 2Department of Arctic and Marine Biology, UiT the Arctic University of Norway, 9037 Tromsø, Norway; laura.jaakola@uit.no; 3Norwegian Institute of Bioeconomy Research (NIBIO), 1431 Ås, Norway; anne.linn.hykkerud@nibio.no (A.L.H.); inger.martinussen@nibio.no (I.M.); 4Food Chemistry and Food Development Unit, Department of Life Technologies, University of Turku, FI-20014 Turku, Finland; niina.m.kelanne@utu.fi (N.K.); saaltu@utu.fi (S.T.); osanla@utu.fi (O.L.); bayang@utu.fi (B.Y.)

**Keywords:** wild berries, cowberry, fruit quality, health, taste, aroma

## Abstract

Wild lingonberries are a traditional source of food in the Nordic countries and an important contributor to economic activity of non-wood forest products in the region. Lingonberries are a rich source of bioactive compounds and can be a valuable contributor to a healthy diet. However, there are few studies available on how the bioactive compounds in lingonberries develop as they ripen. In this investigation, we examined the content of 27 phenolic compounds, three sugars, four organic acids, and 71 volatile organic compounds at five ripening stages. The study showed that, while the highest content of phenolic compounds was found early in the development, the organoleptic quality of the fruits improved as they ripened. From the first to the last stage of development, anthocyanins went from being nearly absent to 100 mg/100 g fw, and there was an increased content of sugars from 2.7 to 7.2 g/100 g fw, whereas the content of organic acids decreased from 4.9 to 2.7 g/100 g fw, and there were several changes in the profile of volatiles. The contents of flavonols, cinnamic acid derivatives, flavan-3-ols, and the total concentration of phenolic compounds were significantly lower in the fully ripe berries compared to berries in the early green stage. In addition to the changes occurring due to ripening, there was observed variation in the profile of both phenolic compounds and volatiles, depending on the growth location of the berries. The present data are useful for the assessment of harvest time to obtain the desired quality of lingonberries.

## 1. Introduction

Lingonberry (*Vaccinium vitis-idaea* L.), belonging to the family of the *Ericaceae* and the genus *Vaccinium*, is a perennial dwarf shrub with scarlet red fruits [1]. The *Vaccinium* genus also contains blueberries (*V. corymbosum*), cranberries (*V. macrocarpon*), and bilberries (*V. myrtillus*), which are increasing in popularity worldwide. Lingonberries grow wildly and abundantly in forests across the Nordic countries and are an important source of income for the Nordic rural communities. Despite an increasing interest in wild berries, only an average of 7.6% of the total yield of lingonberries is harvested annually [2,3]. Lingonberries are mainly sold as fresh or frozen, or produced into jams. These berries are known to have a sour and astringent taste found challenging for many consumers, and appraised the reason for the relatively low consumption of lingonberries [4]. The flavor profile of lingonberries is mostly attributed to both the low content of sugars and the high content of phenolic compounds and organic acids in the berries [5,6,7,8]. Flavour is also affected by volatile organic compounds (VOCs), but little is known about the aroma profile of lingonberries [9]. Lingonberries have a super-food potential, as several studies have found health beneficial effects of including them in the diet [10,11,12,13,14].

The ripening of berries in the *Vaccinium* genus is non-climacteric, with a continuous respiration rate throughout the process. After berry formation, several processes occur simultaneously, causing changes in the composition of the berries. In various *Vaccinium* berries, large changes in the composition of phenolics and terpenoids during the season have been observed [15,16,17,18,19,20]. The synthesis of most phenolic compounds starts at fruit onset, with the highest content of total phenolic compounds observed in immature fruits [16,19,21]. In immature fruits, the content of sugars is low and organic acids is high, and, as the berries ripen, the sugar content increases and organic acid content decreases [22]. As reduced astringency in the berries is considered an important factor for increased use of lingonberries, it is important to study the effects of ripening on composition. The development of volatile compounds has not been extensively studied in lingonberries [9]. In grapes, however, ripening has generally been found to be characterized by accumulation of alcohols, fruity esters, and terpenes during the later stages of development [23]. It is also known that the ripening process is influenced by growth conditions [15,18,24].

The process of berry ripening is visually expressed with an increase in size and accumulation of color in their skin [17,20]. Due to the sour taste and astringency of the berries, it is key to understand how the targeted compounds, influencing the flavor, develop during ripening. Ripening also influences the health properties of lingonberries. The objective of this study was to investigate how sugars, organic acids, and phenolic and volatile organic compounds vary in lingonberries as they ripen. Changes in the metabolic profile during maturation are considered to influence the flavor and antioxidant and antimicrobial potential of the berries. Improved understanding of the ripening processes is essential to increase the use of lingonberries and can be used to optimize harvest time and quality control.

## 2. Materials and Methods

### 2.1. Lingonberry Sample Material

#### 2.1.1. Harvest of Wild Berry Samples

Wild lingonberries were sampled between July and September 2020, from three natural stands (250 m^2^) in southern Norway within a 3 km radius (Appendix A). Whole undamaged lingonberries (100 g) were harvested by hand at five stages of ripening based on predetermined visual criteria of the berry skin surface color, ranging from large unripe green fruit in July to fully ripe fruit in September (Table 1). After harvest, the berries were frozen within three hours and stored at −40 °C until further analysis.

#### 2.1.2. Chemicals

Water used in the experiments was purified by a Milli-Q purification system (Millipore Sigma, Burlington, MA, USA), and solvents used were of HPLC-isocratic grade or higher. The chemicals used as standards in the experiments were purchased from several different vendors. From Acros Organics (Antwerp, Belgium), we purchased 2-hexenal. From Chem Service Inc. (West Chester, PA, USA), we purchased acetophenone, fructose, glucose, and sucrose. From Fluka (Steinheim, Switzerland), we purchased chlorogenic acid and eucalyptol. From Merck (Darmstadt, Germany), we purchased quinic acid. From Polyphenols AS (Sandnes, Norway), we purchased cyanidin-3-O-galactoside. From Sigma-Aldrich (St. Louis, MO, USA), we purchased benzaldehyde, catechin hydrate, citric acid, o-cymene, *p*-cymene, 2-ethyl furan, ethyl octanoate, (*E,E*)-2,4-heptadienal, (*E,E*)-2,4-hexadienal, hexanal, hexanoic acid, 1-hexanol, hexyl acetate, linalool, d-limonene, malic acid, methylbutanal, 3-methyl-1-butanol acetate, 6-methyl-5-hepten-2-one, 4-Methyl-2-pentanol, n-alkane mixture (C7–C30), neryl acetate, 1-octen-3-ol, 1-pentanol, quercetin-3-O-rutinoside, 2-α-pinene, shikimic acid, and *γ*-terpinene.

#### 2.1.3. Methanolic Extraction of Lingonberries

The non-volatile metabolites in lingonberries were extracted in duplicate at ambient temperature (20–22 °C) following the methanolic extraction method described previously [26,27]. Frozen lingonberries (~50 g) were milled before lyophilization for 72 h at a pressure of 0.633 mbar (Gamma 1–16, Christ GmbH, Osterode am Harz, Germany). Dry samples (400 ± 10 mg) were mixed with 5 mL 70% methanol in water (*v*/*v*) in a vortex mixer for 15 s before sonication for 10 min (Ultrasonic Cleaner, VWR International, Radnor, PA, USA) and centrifugation for 10 min at 39,200× *g* (Avanti J-26 XP Centrifuge, Beckman Coulter, Brea, CA, USA). After collection of the supernatant, a re-extraction of the insoluble material was performed. Supernatants were pooled, and the volume was made up to 20 mL with the extraction solvent (70% methanol in water). The extracts were filtered through Millex HA 0.45 μm filters (Millipore Corp., Burlington, MA, USA) before transfer to HPLC vials and stored at −80 °C until analysis.

### 2.2. Analysis of Sugars and Organic Acids

Sugars and organic acid content were determined using an Agilent 1100 series HPLC system, equipped with a diode array detector (DAD) and a refractometer index (RI) detector (Model 132; Gilson, Villiers-le-Bel, France), as previously described by Woznicki et al. [28]. The methanolic extracts (20 μL) were injected in a randomized order, and separation was performed on a Rezex ROA-Organic acid H+ (8%) column (300 × 7.8 mm; Phenomenex, Torrance, CA, USA) at 45 °C with mobile phase 7.2 mmol/L H_2_SO_4_ run at a flow rate of 0.5 mL/min. The detection of the sugars was performed with a RI detector and the organic acid detection was performed with DAD at 210 nm. Identification and quantification was performed according to the method described by Amundsen, et al. [24], and results were presented on a equivalent g/100 g fresh weight (fw) basis.

### 2.3. Analysis of Anthocyanins, Flavonols, Cinnamic Acid Derivatives and Procyanidins

Targeted analysis of phenolic compounds was performed using an Agilent 1100 series HPLC system (Agilent Technologies, Waldbronn, Germany) equipped with an autosampler cooled to 4 °C, a DAD, and a MSD XCT ion trap mass spectrometer was fitted with an electrospray ionization interface with the method described by Aaby, et al. [29]. In a randomized order, 10 μL of the extracts were injected, and separation was performed on a Synergi 4 μm MAX RP C12 column (250 mm × 2.0 mm i.d.), to which a 5 μm C12 guard column (4.0 mm × 2.0 mm i.d.) was equipped. Both columns were produced by Phenomenex (Torrance, CA, USA). The mobile phases used were set up as a binary solvent system, which consisted of: (A) formic acid/water (2/98, *v/v*) and (B) acetonitrile. The two solvents were used in a gradient of: 0–10 min 5–10% B, 10–22 min 10–12.4% B, 22–42 min 12.4–28% B, 42–50 min 28–60% B, 50–55 min 60% B, and 55–58 min 60–5% B. Elution was performed with a flow rate of 0.25 mL/min at 40 °C, with a total run time of 60 min. The mass spectrometer (MS) was operated in positive and negative ion modes according to a previously described method by Aaby et al. [30]. Identification and quantification were performed according to the method described by [24], and results were presented on a equivalent mg/100 g fresh weight (fw) basis.

### 2.4. Analysis of Volatile Organic Compounds

The analysis on the composition of volatile organic compounds (VOCs) was performed according to the method previously described by Marsol-Vall et al. [31]. Determination of the VOC composition was performed with a Trace 1310 gas chromatograph coupled with a TSQ 7000 EVO mass spectrometer (Thermo Scientific, Reinach, Switzerland). To extract volatile compounds, a TriPlus RSH multipurpose autosampler (Thermo Scientific, Reinach, Switzerland) equipped with a HS-SPME with a 2 cm DVB/CAR/PDMS 50/30 µm fiber (Supelco, Bellefonte, PA, USA) was used. Two grams of a lingonberry sample, which was partially thawed, was weighed to a 20 mL headspace vial and spiked with 10 µL of the internal standard mix (4-methyl-2-pentanol at 100 µg/mL and neryl acetate at 113 µg/L in methanol). Then, the berries were crushed. Equilibration of the sample was performed for 10 min at 45 °C, followed by exposure of the fiber to the headspace of the sample vial for 30 min at 45 °C. Separation was performed using a polar capillary column (DB-WAX, 60 m × 0.25 mm × 0.25 µm: J&W Scientific, Folsom, CA, USA). The carrier gas helium was used with a constant flow of 1.6 mL/min. Identification and quantification were performed according to the method described by [24] and expressed as normalized peak areas (compound area/ISTD area). Each lingonberry sample was analyzed in quadruplicate.

### 2.5. Statistical Analysis

To assess the effect of ripening on groups of compounds in lingonberry, a two-way analysis of variance (ANOVA) and a Tukey’s Honestly Significant Difference (HSD) test were performed, reporting significant differences at level *p* < 0.05. To illustrate the variation in the composition of volatile compounds in the samples, a principal component analysis (PCA) was performed with the Unscrambler Software (The Unscrambler^®^X version 10.4.1, CAMO Software AS, Oslo, Norway). Pareto scaling (weighed by 1/square root of the standard deviation) was applied before the multivariate data analysis.

## 3. Results and Discussion

### 3.1. Sugars and Organic Acids

There were three sugars and four organic acids identified in lingonberries in this study: sucrose, glucose and fructose, and quinic, citric, malic, and shikimic acid, respectively. During ripening, the total content of sugars increased from 2.7 g/100 g fw in the early green berries to 7.2 g/100 g fw in the late season ripe berries (Table 2, Appendix A), whereas the content of organic acids decreased from 4.9 g/100 g fw to 2.7 g/100 g fw during the same period. There were no major changes in the profile of sugars during ripening, though the proportion of sucrose (8.8–4.3%) and glucose (50–47%) slightly decreased, whereas the proportion of fructose (42–49%) increased during ripening. Although both glucose and fructose increased in absolute amounts, the sucrose content remained relatively stable throughout ripening. During ripening, particularly the concentrations of quinic and malic acid decreased, whereas only insignificant changes were observed in the concentration of citric acid. The proportion of citric acid thus increased (42–64%), whereas the proportion of quinic (55–34%) and malic acid (3.2–1.8%) decreased. Shikimic acid contributed only to >1% of the total content of lingonberry organic acids. Shikimic acid has been shown to play a role in the synthesis of phenolic compounds [32]. However, no major changes were detected in its content in the course of lingonberry ripening. In a previous study of blueberries, no changes in the proportions of sugars were detected [22], whereas there was a slightly higher content of fructose than glucose, as well as lower content of sucrose in the berries throughout the ripening period in bilberries [33]. In cranberries, the content of quinic acid decreased, and malic acid increased, whereas there was no change in the citric acid content [34]. The total concentrations of sugars and organic acids in ripe berries were comparable to previous findings in lingonberries [5,35,36]. The simultaneous increase in sugar content with a decrease in organic acids is among the most important and characteristic features of fruit and berry ripening and was thus expected [22]. Additionally, earlier studies on lingonberries and bilberries have indicated increasing content of sugars towards the end of ripening [33,37]. The increase in sugar concentration and decrease in the organic acid concentration improve the palatability of the berries. The ratio of sugar to organic acids, which increased from 0.6 to 2.7 during ripening, influences the perceived sweetness of berries and is likely to influence the liking of the lingonberries [4,8]. As lingonberries are generally considered to have a low degree of sweetness, and addition of sugars is often seen as a necessity in products, and berries with a high ratio of sugars to organic acids would be preferred. Harvest of berries later in the season could thus improve quality as the ratio of sugars to organic acids is significantly higher than earlier in the season.

### 3.2. Phenolic Compounds

There were in total 27 phenolic compounds tentatively identified (Appendix A) and quantified in lingonberries during the five stages of ripening (Table 3). Five anthocyanins, eleven flavonols, eight cinnamic acid derivatives, and three flavan-3-ols were identified. The anthocyanins identified were all glycosides of cyanidin. Flavonols tentatively identified were mostly glycosides of quercetin, with small amounts of kaempferol glycosides. Cinnamic acid derivatives (CADs) identified were hexosides of ferulic, caffeic, sinapic and *p*-coumaric acid, a coumaroyl iridoid and chlorogenic acid. Exact quantification of all flavan-3-ols was not possible due to coelution and low molar absorptivity. However, catechin, a B-type dimer, and an A-type dimer could be quantified. Chemical composition of ripe lingonberries has been investigated in several publications, and the composition of anthocyanins, flavonols, and flavan-3-ols were in line to what has been previously reported [7,15,36,38,39,40,41,42].

In the green lingonberries, the accumulation of anthocyanins had just started with 0.4 mg/100 g fw measured. As the berries ripened, accumulation increased, and the content of anthocyanins was 80 mg/100 g fw in ripe berries and increased to 99 mg/100 g fw late in the season (Table 3, Figure 1). All anthocyanins increased with ripening, but the proportion of cyanidin-3-O-galactoside decreased from 92% to 80% in the ripe berries. This was mostly due to a strong increase in the proportions of cyanidin-3-O-arabinoside and cyanidin-3-O-glucoside that were absent in green berries and had a proportion of 13.1% and 5.7%, respectively, in late season berries. In a previous study, the proportion of cyanidin-3-O-galactoside also decreased from 94% to 81% between early berries on 29th of July and late season berries on the 4th of October. In the same study, increases in both the proportion of cyanidin-3-O-arabinoside from 1% to 11% and cyandin-3-O-glucoside from 5% to 6% were detected [15]. The content of the individual compounds is of importance, as studies have shown that the glycosylation influences the absorption of the lingonberry anthocyanins in humans [43]. The anthocyanins of lingonberries accumulate mostly in the skin with little coloration in the fruit flesh [18]. The increase in the anthocyanin content in the fruit skin is the visual expression of ripening [18]. It is thought that plants accumulate anthocyanins to attract seed dispersers, indicating a ripe berry, but they also protect the berry against outside stressors, such as radiation [16,17,18,19,20]. It has been found that birds prefer lingonberries with higher anthocyanin content [1]. Though the process was gradual, most anthocyanin accumulation had taken place as the berry had turned fully red. However, a significantly higher content was detected in the fully ripe late season berries.

While the anthocyanins accumulate throughout the season from 0.4 to 98.5 mg/100 g fw, the content of flavan-3-ols and CADs decreased from 206.0 to 72.6 mg/100 g fw and from 51.9 to 11.3 mg/100 g fw (Table 3). Additionally, the flavonols decreased from 41.2 and 20.1 mg/100 g fw, but their decrease was more gradual throughout the season. The highest total content of phenolic compounds, consisting of flavonols, flavan-3-ols, and CADs, were found in early green berries (Figure 1). Similar results have been found in many fruits and berries, and it is thought that they are a defense mechanism of the plants to protect the immature seeds against predators and herbivores early in the season [18]. In cranberries, the content of flavonols and proanthocyanidins approximately halved as they ripened [44]. In the present study, the profile of flavonols changed during ripening. The content of all flavonols besides kaempferol-3-O-(HMG)-rhamnoside decreased or remained stable during ripening. A previous study reported a decrease in the total concentration of flavonols in lingonberries during ripening, with a particularly large decrease in the content of glucosides and rhamnosides [15]. In the present study, in the early green and half red lingonberries, the proportion of ferulic acid among the CADs was between 55–60%. As the berries ripened, the large decrease in CAD content was mainly due to the decrease in the concentration of a ferulic acid, hexoside. In ripe berries, the proportion of ferulic acid was 8.5%. The high content of ferulic acid in green berries could be linked to the role of ferulic acid in the crosslinks of structural carbohydrates prior to the softening of cell tissues. The content of a coumaric acid, hexoside, remained high throughout ripening. This is probably due to coumaric acid hexoside being a precursor in the cyanidin synthesis [28]. The two dimeric procyanidins measured (A- and B-type procyanidins) and catechin all decreased during ripening. However, there was a stronger decrease in the concentration of the A-type (74%) than the B-type (54%) procyanidin during ripening. Similarly, in a previous study of lingonberries harvested at several timepoints during the season, a relative decrease in the A-type and increase in the B-type proanthocyanidins was observed [15]. Proanthocyanins are among the most interesting health-promoting compounds in lingonberries [10]. The A-type proanthocyanins found in the highest content in the early fruit are less common and have been shown to have the strongest anti-lipid peroxidation activity [45]. This shows that in the early stage, green berries also contain a high content of interesting compounds that could be utilized.

### 3.3. Volatile Compounds

A total of 71 volatile organic compounds (VOCs) were tentatively identified in lingonberries during ripening (Appendix A). The VOCs consisted of twenty aldehydes, nine esters, of which six were acetates, six ketones, eleven alcohols, eight acids, sixteen terpenoids, and one furan. Previously, five studies have been performed analyzing volatile organic compounds in lingonberries [6,24,31,46,47]. Among the studies, only one has analyzed whole crushed berries [24], three have analyzed lingonberry juices [6,31,46], and one analyzed the press residue from juice production [47]. Aldehydes were the most abundant group of VOCs found in the present study (Figure 2). Among aldehydes, many compounds are known to give various aromas, and they most often have green, fatty, or tallow aromas [48]. In previous research of lingonberries, several compounds that were also identified in this study were highlighted as contributors to aroma using GC-olfactometry, such as 2-methylbutanoic acid [46,47], eucalyptol, hexanal, linalool, and methyl benzoate [31]. Diacetyl and 2-methylpropanoate were also identified, but they could not be quantified in this study [31]. Additionally, in cranberries, large variation in the profile of VOC compounds has been found [9]. Most studies of cranberries report the presence of benzyl compounds, including benzyl alcohol, benzoic acid, and benzaldehyde. Additionally, a major part of the studies also have found the terpene α-terpineol in relatively high levels. These compounds were also identified in lingonberries in this study.

During ripening, several changes occurred in the profile of VOCs in lingonberries (Figure 2). During ripening, also, different numbers of compounds were measured: 57 at the first stage of ripening, 63 at the second stage, 67 at the third, 63 at fourth, and 65 at the fifth (Appendix A). The highest total content of VOCs was found in the late season berries closely followed by the red berries. This is in contrast to what has been shown in grapes, in which a higher number of volatiles was found before initiation of ripening (pre- compared to post-veraison fruit) [49]. Previous research has shown that VOCs, amongst others, act as signaling molecules to attract pollinators and seed dispersers, and it is, therefore, natural that a high content is found just prior to ripening. This has been shown in highbush blueberries (*V. corymbosum* L.), in which the highest content of VOCs was found just prior to ripening [50]. In lingonberries, an increase in the content of volatile acids and a tendency for increase in the content of ketones, terpenes, and furan during ripening was detected (Figure 2). Synthesis of volatile compounds is part of other metabolic processes occurring in the berries during ripening. The continuous increase in volatile, short chained fatty acids (C5–C9) during ripening could be due to an increase in the fatty acid fraction in the wax layer of the lingonberry surface during berry ripening [51]. The content of esters was highest in green fruits. When the coloration of the skin had begun, there was little variation in their contents. This is contrary to their role in fully mature blueberries, where esters previously have been proposed to be responsible for fruity aroma characteristics [9]. In lingonberries, a high content of alcohols in the green berries was detected, as well as a tendency for an increase in the late season berries. Two studies of *V. padifolium* showed an increase in the content of alcohols in the ripe berries compared to earlier in the season [19]. Additionally in red grapes, increased alcohol concentrations occur as the berries ripen, whereas the concentration of aldehydes decreased [23]. The content of aldehydes in the present study, however, did not decrease as the berries ripen, as the content was highest in red and late season berries (Figure 2). The high temperature detected when collecting the red berries (Table 1) may have influenced the release of aldehydes and terpenes at this stage. Grapes harvested at higher temperatures have previously been shown to contain a higher content of terpenes [52]. The content of benzyl acetate, benzyl alcohol, and methyl benzoate increased in lingonberries as the berries ripened (Appendix A). This is in line with previous findings showing that the content of benzoic acids increases as the berries ripen [34].

Clear differences in the profiles and responses between the three stands were found, particularly, in the profile of aldehydes and terpenes (Figure 2). Berries from stands A and C were picked from pine forest with medium forest density, whereas the berries from stand B were picked from a clear-cut area. As berries in a clearcut forest are more exposed to radiation than berries grown in a denser forest, this may affect the synthesis of metabolites, such as terpenes. However, lingonberry genetics, growth environment, and their interaction can all influence the synthesis of volatile compounds in a complex manner [18]. In grapes, large variation in aroma compounds has been between different cultivars independent of grape color [23]. Similarly to the results from our study, it has previously been shown in grapes that the terpenes were influenced significantly by UV-B radiation during ripening [52]. This could also explain why the distance from forests has been shown to influence the composition of grapes, which could be due to shading effect [49]. Additionally, genetic background affects the VOC composition, for example, in both highbush and rabbiteye blueberries, large variations have been found between different cultivars during ripening [9]. With the changes observed in the number of VOCs, and their relative quantities, this study indicates that the ripeness of lingonberries affects the profile of VOCs in the berries, which is likely to influence the perceived aroma.

### 3.4. General Discussion

In the PCA of all the measured compounds from the three locations (A, B, C) at the five ripening stages (1–5), the first two components explained 86% of the variation in the data set (Figure 3A). There was a clear clustering of both ripening stage and location. The five ripeness stages spread in the first principal component (73%), showing that the ripeness most significantly influenced the chemical composition of lingonberries. The difference between stands spread in the second principal component (13%) with berries from stand B being separated from berries from stands A and C. Though there was a clear horizontal separation between each ripening stage, it was an evident clustering of berries before (1–2) and after full coloration of the skin (3–5). The cluster of the most ripe berries was characterized by the highest content of anthocyanins, sugars, volatile acids, benzyl alcohol, and acetophenone (Figure 3B). In contrast, the berries prior to ripening were characterized by high content of several of the CADs and flavonols, the flavan-3-ols, malic and quinic acid, methyl acetate, 1-octen-3-ol, 5-methyl-hepthen-one, caryophyllene, and 2,4-heptadienal. Early in the season, lingonberries from a single cluster can vary in ripeness depending on their position. Berries less exposed to sunlight appear to ripen at a slower rate. These differences in ripening on a bush can influence the overall quality of the product. Therefore, efforts should be placed on separation of berries based on color if the berries are harvested early in the season. This study gives clear evidence for a variation in the profile of volatiles in lingonberries with ripening. However, there is still a large gap within the understanding of the influence of changes in composition and their relation to fruit flavor itself. This is in part due to the complex nature of the interactions between the compounds in aroma. Additionally, while lingonberries generally have a high content of phenolic compounds, there is not enough evidence available to determine how variation in these compounds during ripening will influence the berry aroma.

The samples from each of the sampling stands were most clustered in the second and third dimension. Particularly, stand B stood out, but there were differences observed between the five stages of ripening. Though the lingonberries were collected from three locations within a 3 km radius from one another, there was a large variation in the chemical composition particularly in the composition of terpenes and aldehydes between the three locations. This could be due to different forest density, thus different solar radiation, at the three stands, as previously discussed (Section 3.3). The variation in the content of flavonoids is also likely influenced by differences in the growth environment as both radiation and temperature have previously been shown to influence their composition [24,37,53]. However, a recently published paper shows that the composition of both phenolic compounds and triterpenoids in lingonberries grown under the same conditions is strongly influenced by genetic variation of the plants [54]. Plants with different genetic backgrounds may also react differently to different environmental conditions [15,18,24,37,55].

## 4. Conclusions

The chemical composition of lingonberries changed markedly during fruit ripening. The highest contents of phenolic compounds, especially proanthocyanidins, were found in the early stages of berry development, whereas the highest content of anthocyanins and a richer aroma profile was detected in the late harvested lingonberries. Therefore, for the traditional food uses, highest quality berries are achieved by waiting until late in the season with harvest. Potentially, the green berries often harvested together with the first ripe berries in the early season, could be separated and utilized for other purposes. The results reported are of interest for both consumers and producers, as they highlight the influence of ripening and harvest time on the quality of the berries. Further studies are needed for better understanding the relation between volatile compounds and aroma profiles of lingonberries, and the impact of genetic and environmental factors on variation of the aroma profiles and flavor of the berries.

## Figures and Tables

**Figure 1 foods-12-02154-f001:**
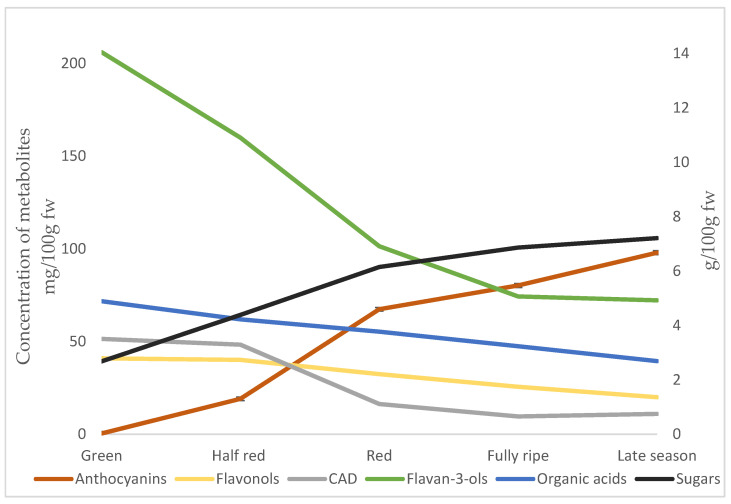
Concentrations of groups of phenolic compounds (mg/100 g fw), sugars, and organic acids (g/100 g fw) in lingonberries at the five different stages of ripening. CAD—cinnamic acid derivatives.

**Figure 2 foods-12-02154-f002:**
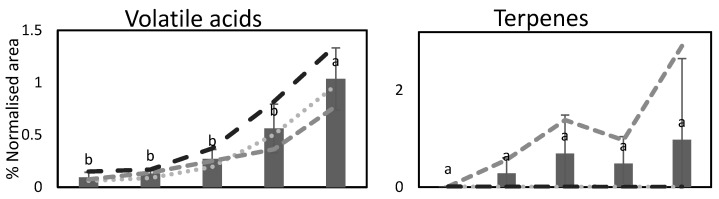
Changes in the mean content of aldehydes, volatile acids, alcohols, ketones, esters, terpenes, and ethyl furan in % normalized area of the internal standard in lingonberries at the three stands (A–C) at the five stages of ripening. Different letters (a–b) indicate significant differences (*p* < 0.05) between the samples, as determined by Tukey’s HSD test.

**Figure 3 foods-12-02154-f003:**
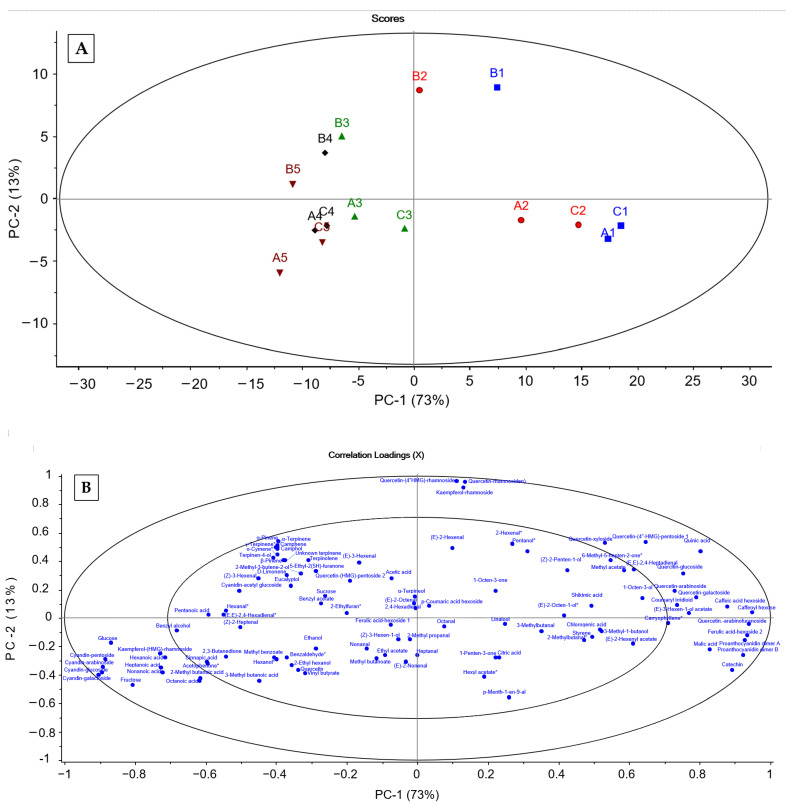
Plots after principal component analysis (PCA) of sugars, organic acids, phenolic compounds, and volatile organic compounds in the lingonberry samples. (**A**) Score plot of berries harvested from three locations (A–C) at five ripening stages; green berries (1; blue squares), half red berries (2; red circles), red but not fully ripe (3; green triangles), ripe berries (4; black diamonds); and late season berries (5; inverted brown triangles). (**B**) Loading plot showing the contribution of each compound in the experiment to the differences between the samples.

**Table 1 foods-12-02154-t001:** Characterisation of the lingonberry samples ^a^ and weather characteristics at the sampling day ^b^.

Harvest Date	Fruit Color	Weight(g/berry)	Dry Weight (%)	Temp Mean (°C)	Max (°C)	Min (°C)	Rain (mm)
23 July 2020	Unripe green	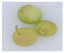	0.21 ± 0.05	15.5 ± 0.2	13.2	19.4	5.0	0
24 July 2020	Half red	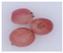	0.23 ± 0.08	15.3 ± 1.1	14.2	20.4	7.8	0
8 August 2020	Ripening fully red	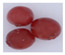	0.28 ± 0.10	15.6 ± 1.2	19.0	25.2	12.4	0.1
27 August 2020	Fully ripe scarlet red	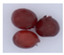	0.28 ± 0.05	15.0 ± 1.5	11.8	19.9	4.2	0
28 September 2020	Late season scarlet red	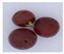	0.27 ± 0.02	15.1 ± 1.2	9.7	15.4	2.5	0.1

^a^ Weight and dry weights are mean values ± standard deviation of three samples. ^b^ Weather data were obtained from the Norwegian Meteorological Institutes Ås weather station and represent mean, minimum, and maximum daily temperature and precipitation during the day of harvest [25].

**Table 2 foods-12-02154-t002:** Concentration of sugars and organic acids in lingonberries picked at five ripening stages ^a^.

	Green	Half Red	Red	Fully Ripe	Late Season
Sucrose	0.23 ± 0.01	0.37 ± 0.07	0.38 ± 0.07	0.34 ± 0.06	0.31 ± 0.02
Fructose	1.1 ± 0.3 d	1.9 ± 0.3 c	2.7 ± 0.5 b	3.2 ± 0.1 ab	3.5 ± 0.4 a
Glucose	1.3 ± 0.2 b	2.2 ± 0.2 b	3.0 ± 0.5 a	3.3 ± 0.4 a	3.4 ± 0.1 a
Total sugars	2.7 ± 0.5 c	4.4 ± 0.4 b	6.2 ± 1.0 a	6.9 ± 0.4 a	7.2 ± 0.4 a
Citric acids	2.1 ± 0.4	2.2 ± 0.3	2.1 ± 0.3	2.1 ± 0.6	1.7 ± 0.4
Quinic acid	2.7 ± 0.5 a	1.9 ± 0.1 b	1.5 ± 0.2 bc	1.1 ± 0.2 cd	0.9 ± 0.1 d
Malic acid	155 ± 39 a	114 ± 54 ab	93 ± 24 abc	79 ± 25 bc	48 ± 17 c
Shikimic acid	3.0 ± 0.7	2.4 ± 0.3	2.3 ± 0.5	2.3 ± 1.0	2.3 ± 0.9
Total organic acids	4.9 ± 0.4 a	4.2 ± 40.3 ab	3.8 ± 0.1 bc	3.2 ± 0.7 cd	2.7 ± 0.5 d
Sugar: acid ratio	0.6 ± 0.2 b	1.0 ± 0.1 b	1.6 ± 0.3 b	2.2 ± 0.4 ab	2.7 ± 0.3 a

^a^ All concentrations are mean values ± standard deviation of three samples analyzed in duplicate presented as concentration in g/100 g fw, except shikimic and malic acid in mg/100 g fw. Different letters (a–d) indicate significant differences (*p* < 0.05) between the samples, as determined by Tukey’s HSD test.

**Table 3 foods-12-02154-t003:** Content of phenolic compounds in lingonberries picked at five ripening stages ^a^.

	Green	Half Red	Red	Fully Ripe	Late Season
Cyanidin-3-O-galactoside	0.4 ± 0.1 b	17.8 ± 1.3 b	57.9 ± 4.0 a	65.4 ± 11.0 a	79.3 ± 17.5 a
Cyanidin-3-O-glucoside	0.0 ± 0.0 c	0.6 ± 0.1 c	3.1 ± 0.5 b	4.5 ± 1.0 ab	5.6 ± 1.5 a
Cyanidin-3-O-arabinoside	0.0 ± 0.0 c	0.9 ± 0.0 c	6.6 ± 0.2 b	10.0 ± 0.7 ab	13.0 ± 3.4 a
Cyanidin-3-O-pentoside	0.0 ± 0.0 c	0.1 ± 0.0 c	0.6 ± 0.1 b	0.9 ± 0.1 ab	1.2 ± 0.4 a
Cyanidin-3-O-(acetyl)glucoside	0.0 ± 0.0	0.0 ± 0.0	0.0 ± 0.1	0.1 ± 0.1	0.1 ± 0.1
Total anthocyanins	0.4 ± 0.1 c	19.4 ± 0.7 c	68.3 ± 2.3 b	80.9 ± 7.0 ab	98.5 ± 12.6 a
Quercetin-3-*O*-galactoside	10.4 ± 2.3 a	10.5 ± 4.1 a	7.6 ± 2.8 ab	5.5 ± 0.6 b	4.6 ± 2.0 b
Quercetin-3-*O*-glucoside	2.4 ± 0.2 a	2.3 ± 0.4 a	1.6 ± 0.3 ab	1.2 ± 0.1 b	1.2 ± 0.3 b
Quercetin-3-*O*-xyloside	1.7 ± 0.2	1.6 ± 0.3	1.4 ± 0.1	1.3 ± 0.2	1.2 ± 0.2
Quercetin-3-*O*-arabinoside	10.4 ± 0.8 a	10.8 ± 0.6 a	9.5 ± 1.3 ab	7.8 ± 0.5 bc	6.6 ± 0.8 c
Quercetin-3-*O*-arabinofuranoside	1.0 ± 0.1 a	0.8 ± 0.1 b	0.6 ± 0.1 c	0.5 ± 0.1 d	0.4 ± 0.1 d
Quercetin-3-*O*-rhamnoside	6.3 ± 4.5	6.3 ± 4.8	5.2 ± 3.6	3.8 ± 2.6	2.4 ± 1.7
Quercetin-(HMG)-pentoside	0.4 ± 0.1 a	0.4 ± 0.1 ab	0.3 ± 0.0 abc	0.3 ± 0.0 bc	0.2 ± 0.0 c
Quercetin-3-*O*-(HMG)-pentoside 2 ^b^	0.0 ± 0.0 b	0.1 ± 0.1 ab	0.2 ± 0.1 a	0.0 ± 0.1 b	0.0 ± 0.0 b
Kaempferol-3-O-rhamnoside	0.4 ± 0.2 a	0.3 ± 0.2 ab	0.3 ± 0.1 ab	0.2 ± 0.1 ab	0.2 ± 0.0 b
Quercetin-3-*O*-(HMG)-rhamnoside ^b^	8.2 ± 7.8	7.6 ± 5.8	5.7 ± 4.9	4.6 ± 4.3	2.8 ± 2.6
Kaempferol-3-*O*-(HMG)-rhamnoside ^b^	0.0 ± 0.0 b	0.1 ± 0.0 b	0.4 ± 0.1 ab	0.7 ± 0.2 b	0.5 ± 0.1 b
Total flavonols	41.2 ± 7.7 a	40.8 ± 7.3 a	32.8 ± 5.3 ab	25.8 ± 4.2 b	20.1 ± 1.2 b
Ferulic acid-hexoside 1	1.2 ± 0.8 c	2.4 ± 0.4 ab	2.8 ± 0.6 a	1.5 ± 0.2 bc	1.3 ± 0.2 c
Ferulic acid-hexoside 2	31.4 ± 11.2 a	30 ± 19.2 a	3.6 ± 1.5 b	0.8 ± 0.1 b	0.7 ± 0.1 b
Coumaroyl iridoid	5.7 ± 2.2 a	2.9 ± 1.4 ab	1.6 ± 0.8 b	1.1 ± 0.8 b	1.0 ± 0.8 b
Caffeic acid hexoside 1	2.1 ± 0.3 a	1.9 ± 0.5 ab	1.1 ± 0.1 bc	0.8 ± 0.2 c	0.7 ± 0.1 c
Caffeic acid hexoside 2	3.1 ± 0.3 a	2.9 ± 1.6 a	1.0 ± 0.3 b	0.5 ± 0.1 b	0.3 ± 0.0 b
*p*-Coumaric acid hexoside	5.7 ± 0.5	4.7 ± 1.1	5.3 ± 0.1	4.2 ± 0.3	5.7 ± 1.6
Chlorogenic acid	2.6 ± 0.6	3.8 ± 4.4	1.8 ± 1.0	1.3 ± 0.1	1.7 ± 0.7
Sinapic acid hexoside	0.1 ± 0.0 a	0.1 ± 0.0 ab	0.1 ± 0.0 ab	0.1 ± 0.0 b	0.1 ± 0.0 b
Total cinnamic acid derivatives	51.9 ± 6.4 a	48.8 ± 14.1 a	17.4 ± 1.7 b	10.3 ± 0.7 b	11.3 ± 1.1 b
Proanthocyanidin dimer A	103.8 ± 41.7 a	78.8 ± 34.6 ab	46.8 ± 28.3 b	34.1 ± 13.6 b	27.2 ± 4.3 b
Proanthocyanidin dimer B	30.8 ± 7.4 a	26.7 ± 8.3 ab	17.8 ± 3.5 bc	14.1 ± 3.5 c	14.2 ± 4.7 c
Catechin	71.4 ± 24.6 a	54.4 ± 20.3 ab	36.7 ± 10.7 bc	26.1 ± 7.9 c	31.4 ± 11.4 c
Total Flavan-3-ols	206.0 ± 67.8 a	159.8 ± 61 ab	101.4 ± 33.4 bc	74.3 ± 10.7 c	72.8 ± 14.6 c

^a^ Concentrations are mean values ± standard deviation of three samples analyzed in duplicate. Anthocyanins were quantified as mg/100 g fresh weight (fw) equivalents of cyanidin-3-galactoside at 520 nm, flavonol glycosides as quercetin-3-rutinoside at 360 nm, cinnamic acid derivatives as chlorogenic acid at 320 nm, and flavan-3-ols as catechin at 280 nm. Different letters (a–d) indicate significant differences (*p* < 0.05) between the samples, as determined by Tukey’s HSD test. ^b^ HMG = Hydroxy-3-methylglutaroyl.

## Data Availability

Data is contained within the article or Appendix A.

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
