# Peer review of "Composition of Sugars, Organic Acids, Phenolic Compounds, and Volatile Organic Compounds in Lingonberries (Vaccinium vitis-idaea L.) at Five Ripening Stages"

_foods, 2023, doi:10.3390/foods12112154_

Round 1

Reviewer 1 Report

Not clear why shikimic acid was included in the analysis, as its importance to lingonberries and the analyzed results (shikimic acid levels did not change and were a small percentage of total acids) were not discussed.

The quality of the English is high (way better than my Norwegian - skal!), but more idiosyncratic uses of English showed up in discussions of the results.

line 78 - word missing before "analysis"?

line 128 - "DAD" abbreviation should be inserted at first use of "diode array detector" on line 121

line 156 - "Initially" is confusing for this reader, who expected that a change was made in the program; what happened secondarily?

line 199 - suggest you add "period" after "vegetation"

line 250 - replace "a" with a comma

line 265 - "seen" is superfluous, and can be deleted

line 270 - delete comma

line 272 - unclear what "This decrease" refers to, as the previous sentence is about the proportion of ferulic acid in unripe lingonberries

line 302 - is "compounds were" missing between "These" and "also"?

line 311 - "and the and the furan" is confusing

line 315 - "there were" can be deleted

line 317-318 - it might be rewritten as "where before initiation... a higher number of volatiles was found than postveraison."

line 325 - is there a particular furan of interest? if not, I suggest that you delete "the"

Please check sentences starting on lines 328, 343, 344 - seems like a word or two are missing

line 351 - "for" can be deleted; this paragraph mixes sentences about grapes with sentences about lingonberries, and your thoughts about how stand characteristics affect the chemistry of lingonberries might be clearer to readers if first grape research findings are discussed, followed by how lingonberries are similar to or different from grapes

line 371 - delete "and"?

line 382 - substitute "In contrast" for "Whereas"?

lines 387-389 - meaning of "as they are often found on the bottom" unclear to this reader; are you trying to convey the idea that some berries are hidden in a cluster and therefore ripen later than more exposed berries?

line 391 - unclear what "its" refers to - variation in volatile profile? the changes (in what)?

line 412 - sentence could be clearer; do you mean that stand characteristics, most likely radiation differences, affected the profile and variation in flavonoids?

line 421 - substitute "until" for "to"

lines 422-423 - seems awkward; maybe rewrite as "...producers, to identify when berries should be picked to ensure a tasty, healthy, high-quality optimal product."

Author Response

Comments and Suggestions for Authors

Not clear why shikimic acid was included in the analysis, as its importance to lingonberries and the analyzed results (shikimic acid levels did not change and were a small percentage of total acids) were not discussed.

While it is true that shikimic acid did not contribute greatly to the overall content of organic acids, it was considered that it should be reported nevertheless as a part of the compounds identified..The fact that there was no significant difference  was also the reason why it was not extensively discussed. However, as it would give some better explanation to why it was included, a short discussion of its relevance has been added (lines 274-277).

Comments on the Quality of English Language

The quality of the English is high (way better than my Norwegian - skal!), but more idiosyncratic uses of English showed up in discussions of the results.

The authors would like to thank reviewer 1 for his/her comments. The comments have been numbered and those numbered 1-21 were all considered to improve the text and the suggested changes were all been implemented.

  1. line 78 - word missing before "analysis"?
  2. line 128 - "DAD" abbreviation should be inserted at first use of "diode array detector" on line 121
  3. line 156 - "Initially" is confusing for this reader, who expected that a change was made in the program; what happened secondarily?
  4. line 199 - suggest you add "period" after "vegetation"
  5. line 250 - replace "a" with a comma
  6. line 265 - "seen" is superfluous, and can be deleted
  7. line 270 - delete comma
  8. line 272 - unclear what "This decrease" refers to, as the previous sentence is about the proportion of ferulic acid in unripe lingonberries
  9. line 302 - is "compounds were" missing between "These" and "also"?
  10. line 311 - "and the and the furan" is confusing
  11. line 315 - "there were" can be deleted
  12. line 317-318 - it might be rewritten as "where before initiation... a higher number of volatiles was found than postveraison."
  13. line 325 - is there a particular furan of interest? if not, I suggest that you delete "the"
  14. Please check sentences starting on lines 328, 343, 344 - seems like a word or two are missing
  15. line 351 - "for" can be deleted; this paragraph mixes sentences about grapes with sentences about lingonberries, and your thoughts about how stand characteristics affect the chemistry of lingonberries might be clearer to readers if first grape research findings are discussed, followed by how lingonberries are similar to or different from grapes
  16. line 382 - substitute "In contrast" for "Whereas"?
  17. line 391 - unclear what "its" refers to - variation in volatile profile? the changes (in what)?
  18. line 421 - substitute "until" for "to"
  19. line 412 - sentence could be clearer; do you mean that stand characteristics, most likely radiation differences, affected the profile and variation in flavonoids?
    • The paragraph has been revised to better convey the message that a variation of stand characteristics, including radiation differences, affected the profile and variations in flavonoids.
  20. lines 387-389 - meaning of "as they are often found on the bottom" unclear to this reader; are you trying to convey the idea that some berries are hidden in a cluster and therefore ripen later than more exposed berries?
    • The general idea of the argument is that berries on a cluster ripen at different rates depending on their exposure to light, the sentence has been modified to better convey this message in lines 481-483
  21. lines 422-423 - seems awkward; maybe rewrite as "...producers, to identify when berries should be picked to ensure a tasty, healthy, high-quality optimal product."
    • The concluding paragraph has been altered to better reflect the changes made in abstract and introduction in lines 494-596.

Comment numbered 22, was however not considered pertinent as it made the sentence incomplete.

  1. line 371 - delete "and"?

Reviewer 2 Report

The manuscript entitled “Composition of sugars, organic acids, phenolic compounds and volatile organic compounds in lingonberries (Vaccinium Vitis-idaea L.) at five growth stages” deals with the changes of bioactive compounds in lingonberry fruit during ripening.

The topic is very actual in the field of food chemistry, thus fitting the Foods scope, but the novelty of the study is arguable - it mostly confirms existing knowledge of metabolites content in lingonberry fruit.

The major concern is that it is not clearly explain (especially in Introduction and Discussion section) why is important to investigate the content of metabolites in the early stage of fruit ripening, apart from physiological aspect, given that “these products are known to have a sour and astringent taste found difficult to many consumers”, even when they are fully ripe.

The name of compounds should be carefully checked throughout the text, including tables and figures. Occasionally, compound names are written in capital letters, without any reason. Also, sometimes names in text are not linked to names given in tables, thus they should be uniformed.

In text there are sentences that seemed too long and confusing, which should be reformulated or divided into shorter ones to make them clearer.

For the values in the tables, care must be taken about the number of significant digits. It should be uniformed in all tables.

Specific comments are given in the pdf file attached.

In the text there are sentences that seemed too long and confusing, which interrupts the continuity of the text. So, they should be reformulated or divided into shorter ones to make them clearer.

Author Response

Comments and Suggestions for Authors

The manuscript entitled “Composition of sugars, organic acids, phenolic compounds and volatile organic compounds in lingonberries (Vaccinium Vitis-idaea L.) at five growth stages” deals with the changes of bioactive compounds in lingonberry fruit during ripening. The topic is very actual in the field of food chemistry, thus fitting the Foods scope, but the novelty of the study is arguable - it mostly confirms existing knowledge of metabolites content in lingonberry fruit. The major concern is that it is not clearly explain (especially in Introduction and Discussion section) why is important to investigate the content of metabolites in the early stage of fruit ripening, apart from physiological aspect, given that “these products are known to have a sour and astringent taste found difficult to many consumers”, even when they are fully ripe.

It is considered that the detailed analysis of the chemical composition during the ripening process is of  high importance due to the sour and astringent taste found of lingonberries. Several of the compounds studied are influencing these characteristics, and our results show that ripening gave substantial changes. To our knowledge, volatiles have not been described in ripening lingonberries previously. A clarification on the importance of studying the lingonberry metabolites during ripening has now been elaborated in the abstract, introduction and concluding paragraph of the discussion lines 494-506

The name of compounds should be carefully checked throughout the text, including tables and figures. Occasionally, compound names are written in capital letters, without any reason. Also, sometimes names in text are not linked to names given in tables, thus they should be uniformed. In text there are sentences that seemed too long and confusing, which should be reformulated or divided into shorter ones to make them clearer. For the values in the tables, care must be taken about the number of significant digits. It should be uniformed in all tables. Specific comments are given in the pdf file attached.

The manuscript has been revised in detailed with regards to its readability as requested by the reviewer. To ensure accuracy, and to address any missing words and incomplete sentences, all authors have carefully reviewed the final version of the manuscript.  Furthermore, the names of the compounds, significant digits and sentence structures have been updated in the revised manuscript.  

Comments on the Quality of English Language

In the text there are sentences that seemed too long and confusing, which interrupts the continuity of the text. So, they should be reformulated or divided into shorter ones to make them clearer.

See previous comment.

Reviewer 3 Report

Warm congratulations on completing a tremendous amount of work. This achievement deserves to be acknowledged unequivocally. Overall, the entirety is very valuable, but the worst part of it, which I consider to require improvement, is the abstract and introduction. Setting aside the content itself, why are the terms "factors" used in the abstract when they never appear again in the text? The methodologies are written well, although I notice differences in their level of detail. Some of them require more detail (e.g. 2.2). I recommend also improving the quality of Fig. 2. 

As a non-native speaker, it is difficult for me to identify specific language errors. However, as someone who reads many scientific articles in English, I notice difficulties in properly understanding yours.

Author Response

Comments and Suggestions for Authors

Warm congratulations on completing a tremendous amount of work. This achievement deserves to be acknowledged unequivocally. Overall, the entirety is very valuable, but the worst part of it, which I consider to require improvement, is the abstract and introduction. Setting aside the content itself, why are the terms "factors" used in the abstract when they never appear again in the text?

This comment brings up several of the same issues as reviewer 2. To improve the quality of the manuscript and to emphasise the importance of the present research substantial changes have now been made to the entire manuscript with emphasis on the abstract and the introduction.

The methodologies are written well, although I notice differences in their level of detail. Some of them require more detail (e.g. 2.2).

The methodology section has been adapted to take into account the suggestion. However, it is considered that varying level of details in the different sections is needed due to the varying complexity and novelty of the different methods used. Analysis of sugars in section 2.2. has previously been extensively described and the methodology is well established. However, the use of HS-SPME-GC-MS is more novel and requires a higher level of details as the results can be influenced by methodological changes. Therefore, it was considered that more details were needed in the description of this method.

I recommend also improving the quality of Fig. 2. 

Quality of Fig-2. Has been improved.

Comments on the Quality of English Language

As a non-native speaker, it is difficult for me to identify specific language errors. However, as someone who reads many scientific articles in English, I notice difficulties in properly understanding yours.

Reviewer 4 Report

This manuscript aimed to understand how the bioactive compounds in lingonberries develop as they ripen. Changes in the contents of sugars, organic acids, phenolic compounds and volatile organic compounds in lingonberries at five growth stages were studied. Several analyses were carried out. And the results were discussed in detail, while the following issues need to be addressed.

 Additional comments:

1.         Line 78, ‘stored in the freezer at -40 °C analysis’

Did the authors mean ‘stored in a freezer at -40 °C for further analysis’?

2.         For analysis of sugar, organic acid, phenolic compounds, volatile compounds etc., how the samples were treated before analysis? Please clarify. The sample preparation method might have impact on the results of these compounds.

3.         Figure 1, please clarify how the results were obtained. Please label what the y axis is. How many replicated analysis were carried out?

4.         In the results and discussion section, some of the results were only generally described without quantitative comparisons, such as 259-267. It might be more convincing comparing the changes during ripening quantitatively.

5.         Lien 290-292, ‘Five studies were identified having been performed on lingonberries [6,24,34,46,47]. Among the studies only one has analyzed whole crushed berries, three have analyzed lingonberry juices and one analyzed the press residue from juice production.’

What’s the objective of stating this? What are the differences between the previous studies and the present one?

6.         Figure 2, what were the objectives of presenting the results of the three stands (line A-C)? There are grammar and typo issues in the figure caption.

7.         Line 328-329, please check grammar.

8.         It is suggested to add a brief conclusion at the end of this manuscript.

Author Response

Comments and Suggestions for Authors

This manuscript aimed to understand how the bioactive compounds in lingonberries develop as they ripen. Changes in the contents of sugars, organic acids, phenolic compounds and volatile organic compounds in lingonberries at five growth stages were studied. Several analyses were carried out. And the results were discussed in detail, while the following issues need to be addressed.

Comments on the Quality of English Language

  1. Line 78, ‘stored in the freezer at -40 °C analysis’

Did the authors mean ‘stored in a freezer at -40 °C for further analysis’?

The text has been altered to agree with the suggestion of the reviewer line 95.

  1. For analysis of sugar, organic acid, phenolic compounds, volatile compounds etc., how the samples were treated before analysis? Please clarify. The sample preparation method might have impact on the results of these compounds.

       As section 2.1.2. describes the sample preparation method for analyses of sugar, organic acid, phenolic compounds and the sample preparation method for volatile compounds is described in lines 177-182. It is considered that the sample treatment is adequately described. The methods for phenolic compounds have been used on a previous paper on strawberries (Davik et al., 2020) and has been tested to give good results for simultaneous extraction of the target compounds in lingonberries (Aaby and Amundsen, 2021).

  1. Figure 1, please clarify how the results were obtained. Please label what the y axis is. How many replicated analysis were carried out?

The y label has been labeled. The analyses were carried out in duplicate on berries from three stands on the results presented in figure1. For analyses of VOCs four replicated analyses were performed on each of the three stands.

  1. In the results and discussion section, some of the results were only generally described without quantitative comparisons, such as 259-267. It might be more convincing comparing the changes during ripening quantitatively.

To support the discussions, quantitative values have been added to several points in the qualitative discussion.

  1. Lien 290-292, ‘Five studies were identified having been performed on lingonberries [6,24,34,46,47]. Among the studies only one has analyzed whole crushed berries, three have analyzed lingonberry juices and one analyzed the press residue from juice production.’ What’s the objective of stating this? What are the differences between the previous studies and the present one?

As the reviewer states in a previous comment, it is recognized that the pretreatment of the samples has an impact on the results. Analyses of volatile organic compounds are particularly influenced by the differences, and the objective of stating this was to highlight the different pretreatments of the samples. References have been added to the appropriate lines, to improve the feasibility for other readers to find the correct studies.

  1. Figure 2, what were the objectives of presenting the results of the three stands (line A-C)? There are grammar and typo issues in the figure caption.

The aim of presenting the three stands was to capture the differences seen, particularly in aldehydes and terpenes as discussed in (lines 426-432). These differences were considered to be due to genetic or environmental differences between the stands and it was considered that adding the individual stand values, as a means of four replicates, would give a picture of the differences between the stands. Typos have been corrected in the figure caption.

  1. Line 328-329, please check grammar.

Grammar has been checked, and the sentence has been changed accordingly throughout the body of the text.

  1. It is suggested to add a brief conclusion at the end of this manuscript.

A conclusion has been added to the end of the manuscript.

Round 2

Reviewer 2 Report

Dear Authors,

the manuscript is significantly improved, but still some slight corrections are needed, as well as minor editing of English.

Summary of comments are given in pdf file attached.

Minor editing of English is needed.

Author Response

The authors would like to thank the reviewer the comments made, and we have made appropriate changes to the manuscript.  

Reviewer 4 Report

The authors made several improvements after revision, still some issues are remaining. I don't think it could be considered for publication before revision.

More comments are as follows:

1. The ordinated scale was not showed in Figure 1, furthermore, the data of phenolic compounds and sugars, and organic acids in Figure 1 have been showed in Table 2 and Table 3, respectively. it needs to revise.

2. The authors should response the reviewer's comments point to point.

Author Response

The ordinated scale was not showed in Figure 1, furthermore, the data of phenolic compounds and sugars, and organic acids in Figure 1 have been showed in Table 2 and Table 3, respectively. it needs to revise.

The ordinated scales have been added to Figure 1

While it is true that data for Figure 1 are drawn from tables 2 and 3, we consider consider that a combined figure with all the metabolites provides provided a good overview of the developments of the metabolites during ripening and makes it easier for the reader to compare and understand the results. 

2. The authors should response the reviewer's comments point to point.
It is not clear what the reviewer means, as we, to our best knowledge, responded to the comments of the reviewers point to point in round 1.